

# Non-inferiority of retrospective data collection for assessing perioperative morbidity

Amour B.U. Patel[1,2], Anna Reyes[1,2] and Gareth L. Ackland[1,2,3]

[1] Department of Anaesthesia, University College London Hospitals NHS Trust, London, United Kingdom
[2] UCL/UCLH National Institute for Health Research Comprehensive Biomedical Research Centre, University College London Hospitals NHS Trust, London, United Kingdom
[3] William Harvey Research Institute, QMUL Queen Mary, University of London, United Kingdom

Corresponding author
Gareth L. Ackland,
g.ackland@qmul.ac.uk

## ABSTRACT

**Background.** Postoperative morbidity has immediate and delayed consequences for surgical patients, including excess risk of premature death. Capturing these data objectively and routinely in large electronic databases using tools such as the Postoperative Morbidity Survey (POMS) would offer tremendous clinical and translational potential. However, POMS has thus far only utilised prospective data collection by research staff. We hypothesised that retrospective data collection from routinely collated hospital data from paper and electronic charts, medical and nursing notes was non-inferior to prospective data collection requiring research staff capturing POMS-defined morbidity in real-time.

**Methods.** Morbidity was recorded by a trained investigator as defined by POMS prospectively on postoperative days 3 and 7. Separately, an independent investigator blinded to prospectively acquired data retrospectively assessed the same patients' morbidity as defined by POMS criteria, using medical charts, nursing summaries and electronic data. Equivalence was accepted when the confidence limits for both modes of data collection fell completely inside the equivalence bounds, with the maximum equivalence difference (i.e., the largest value of the difference in sensitivities deemed to reach a conclusion of equivalence) set a priori at 0.2. Differences for confidence limits between retrospective and prospective data collection were based on Nam's RMLE method. The relationship between morbidity on postoperative day 3 as recorded by each data collection method on time to become morbidity free and length of hospital stay was compared using the log-rank test.

**Results.** POMS data from 85 patients undergoing elective or emergency surgery were analyzed. At postoperative day 3, POMS-defined morbidity was similar regardless of whether data were collected prospectively or retrospectively (95% CI [−0.13–0.013]; $p < 0.001$). Non-inferiority for sensitivity was observed for all other POMS domains and timepoints. Time to become morbidity free Kaplan–Meier plots were indistinguishable between POMS obtained prospectively or retrospectively (hazard ratio: 1.09 (95% CI [0.76–1.57]); $p = 0.33$, log rank test). Similarly, the mode of data collection did not alter the association between early postoperative morbidity on postoperative day 3 and delayed hospital discharge.

**Conclusions.** Postoperative morbidity as defined by the Post Operative Morbidity Survey can be assessed retrospectively. These data may therefore be easily captured using electronic patient record systems, thereby expanding the potential for bioinformatics approaches to generate new clinical and translational insights into recovery from surgery.

## INTRODUCTION

The development of even minor postoperative complications is a major determinant of hospital readmission, long-term adverse outcomes and death (*Khuri et al., 2005*; *Moonesinghe et al., 2014*). Postoperative morbidity can be recorded using a number of tools, but POMS (Post Operative Morbidity Survey) has emerged as a useful survey for assessing short-term morbidity following moderate-major surgery in clinical (*Bennett-Guerrero et al., 1999*; *Grocott et al., 2007*; *Ackland et al., 2010*; *Ackland et al., 2015*; *Ackland et al., 2011*; *Ackland et al., 2007*; *Ausania et al., 2012*; *Davies et al., 2013*; *Jones et al., 2013*; *Pearse et al., 2014*; *Sanders et al., 2012*; *Snowden et al., 2010*; *Wakeling et al., 2005*) and translational perioperative studies (*Edwards et al., in press*). However, POMS has thus far only utilized prospective data collection, requiring research staff to record morbid events. The potential for electronic capture of these data is under-explored. However, determining whether retrospective, rather than prospective, data collection can capture POMS-defined morbidity is a first step that may help exploit these data for enhanced, large scale bioinformatic interrogation. We hypothesized that retrospective data collection from charts, medical and nursing notes was non-inferior to prospective data collection for capturing POMS-defined morbidity. We tested this by three different approaches. First, we established whether POMS- defined morbidity captured retrospectively was statistically non-inferior (*Walker & Nowacki, 2011*) to prospective, real time data collection, by calculating differences based on Nam's RMLE method (*Nam, 1997*). Second, we assessed whether POMS-defined morbidity captured retrospectively or prospectively altered the trajectory of patients becoming free of postoperative morbidity. Third, we assessed whether POMS-defined morbidity on postoperative day 3 captured retrospectively or prospectively was linked with delayed hospital discharge, the predictive value for which has been established in previous studies (*Ackland et al., 2011*).

## MATERIALS AND METHODS

We analysed POMS in 85 patients undergoing major elective surgery at University College London Hospital, having obtained written informed consent (institutional board review-Medical Research Ethics Committee: 10/WNo03/25). POMS domains are detailed in

**Table 1 POMS-defined morbidity.**

| Morbidity type | |
|---|---|
| Pulmonary | *De novo* requirement for supplemental oxygen or other respiratory support (e.g., continuous positive airway pressure or mechanical ventilation). |
| Infectious | Currently on antibiotics or temperature >38 °C in the last 24 h. |
| Renal | Presence of oliguria (<500 mL/day), increased serum creatinine (>30% from baseline value), or urinary catheter in place for a non-surgical reasons. |
| Gastrointestinal | Unable to tolerate an enteral diet (either by mouth or feeding tube) for any reason, including nausea, vomiting and abdominal distension. |
| Cardiovascular | Diagnostic test or therapy in last 24 h for any of the following reasons: *de novo* myocardial infarction or ischemia, hypotension (requiring drug therapy or fluid >200 mL/h), atrial or ventricular arrhythmia or pulmonary edema. |
| Neurological | Presence of a *de novo* focal deficit, coma or confusion/delirium. |
| Wound | Wound dehiscence requiring surgical exploration or drainage or pus from the wound. |
| Hematological | Requirement for any of the following within last 24 h: blood, platelets, fresh frozen plasma or cryoprecipitate. |
| Pain | Surgical wound pain significant enough to require parenteral opiates or regional anesthesia. |

Table 1. Morbidity was recorded as defined by POMS prospectively on postoperative days 3 and 7. Both investigators were trained in prospectively collecting POMS data at the bedside. Thereafter, one was assigned to prospective data collection at the bedside, while the other assessed POMS from the same patients by retrospective analysis of medical charts, nursing summaries and electronic data blinded to prospectively acquired data. The primary endpoint were whether retrospective data collection demonstrated non-inferiority for sensitivity, compared to data recorded prospectively for all-cause morbidity on postoperative day 3.

## Statistical analysis

Differences for confidence limits between retrospective and prospective data collection were based on Nam's RMLE method (*Nam, 1997*). Maximum equivalence difference, the largest value of the difference in sensitivities deemed to reach a conclusion of equivalence, was set a priori at 0.2 (*Walker & Nowacki, 2011*). Equivalence was accepted when the

confidence limits for both modes of data collection fell completely inside the equivalence bounds. Alpha for testing the hypothesis was set at 0.05. Data are reported as mean ± SD, or confidence limits based on Blackwelder and Nam's method. Time to become morbidity free was compared using the log-rank test. The impact of morbidity on postoperative day 3 as defined by each data collection method on length of hospital stay was compared using the log-rank test. *P* values < 0.05 were considered significant. NCSS 8 (NCSS, LLC., Kaysville, Utah, USA) was used for all statistical analyses.

### Power calculation

We calculated the sample size required to be 89 subjects, to achieve 80% power at a significance level of 0.05 using a one-sided non-inferiority test of correlated proportions when the standard proportion was 0.6. The maximum ratio of these proportions that resulted in non-inferiority (the range of non-inferiority) was set at 0.85, and the actual ratio of the proportions was 1 (PASS 14 Power Analysis and Sample Size Software (2015), NCSS, LLC., Kaysville, Utah, USA).

## RESULTS

The mean age of the cohort was 62 ± 9 y; 46% of patients were male. Eight colorectal, 22 vascular, 29 orthopaedic and 26 urological surgical procedures were performed, lasting 2.4 ± 1.4 h. There were no deaths during the hospital admission.

POMS-defined morbidity at any postoperative time point was identified in 52/85 (61%) of patients by prospective evaluation. POMS-defined morbidity was more common on postoperative day 3 than 7 (Fig. 1). Retrospective analysis similarly identified postoperative morbidity on both postoperative days 3 and 7, with no domains beyond the upper bound confidence limit for non-inferiority (Tables 2 and 3).

### Serial morbidity patterns

Time to become morbidity free analysis (Fig. 1) were indistinguishable based on data collection by retrospective and prospective modes (hazard ratio: 1.09 (95% CI [0.76–1.57]); $p = 0.33$, by log rank test).

### Length of hospital stay

We have previously shown that the presence of POMS-defined morbidity on postoperative day 3 is associated with prolonged hospital stay (*Ackland et al., 2011*). Both retrospective and prospective modes of data collection that identified POMS-defined morbidity on postoperative day 3 showed similar relationships with length of stay (Fig. 2).

## DISCUSSION

This study shows that in patients undergoing elective or emergency surgery, retrospective compilation of POMS-defined morbidity data from charts, medical and nursing notes is non-inferior to prospective, real-time data collection requiring research or administrative personnel. Our data demonstrate that similar proportions of patients were free of postoperative morbidity at a given time-point through data captured retrospectively,

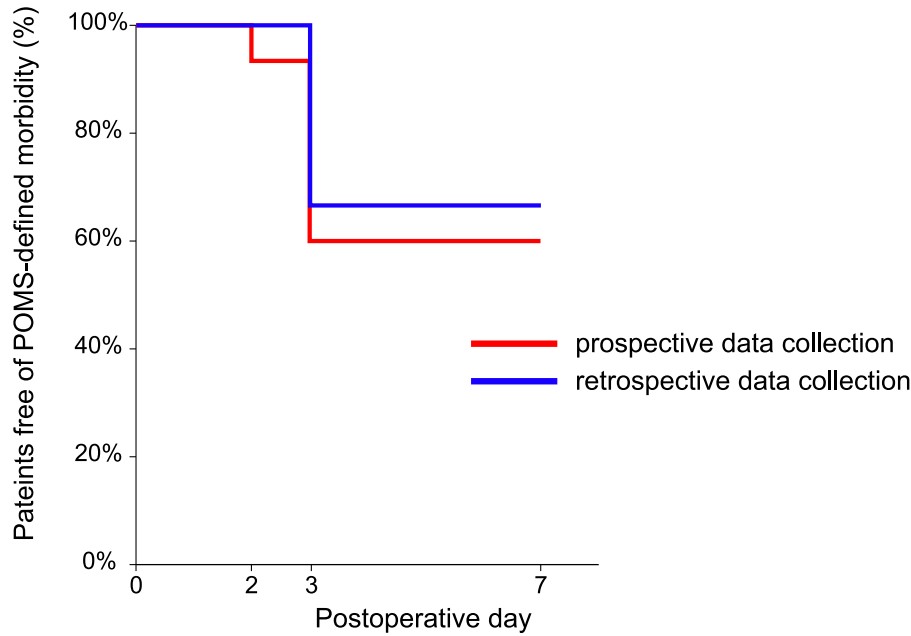

**Figure 1** **Kaplan–Meier plot of percentage of patients free of POMS-defined morbidity against time by postoperative day.** The plot demonstrates a similar percentage of patients were free of postoperative morbidity at a given time-point through data collected retrospectively compared to POMS data collected prospectively (hazard ratio: 1.09 (95% CI [0.76–1.57]); $p = 0.33$, by log rank test).

**Table 2** Postoperative Day 3, equivalence and non-inferiority for sensitivity of mode of data collection.

| Domain | P value | 95% CI | Non-inferior? |
|---|---|---|---|
| Any type | <0.001 | −0.13–0.013 | Yes |
| Supplementary oxygen | <0.001 | −0.24–0.036 | Yes |
| Antibiotics | <0.001 | −0.18–0.0039 | Yes |
| Temperature | <0.001 | −0.058–0.0583 | Yes |
| Urinary catheter | <0.001 | −0.086–0.0432 | Yes |
| PONV | <0.001 | −0.14–0.0309 | Yes |
| Enteral feeding | <0.001 | −0.058–0.0583 | Yes |
| Confusion | <0.001 | −0.058–0.0583 | Yes |
| Myocardial infarction | <0.001 | −0.058–0.0583 | Yes |
| Arrhythmias | <0.001 | −0.058–0.0583 | Yes |
| Drain | <0.001 | −0.058–0.0583 | Yes |
| Packed red cells | <0.001 | −0.11–0.058 | Yes |
| Products | 0.0001 | −0.018–0.072 | Yes |
| Pain | 0.0001 | −0.018–0.072 | Yes |
| Parenteral opioids | <0.001 | −0.2025–0.01 | Yes |

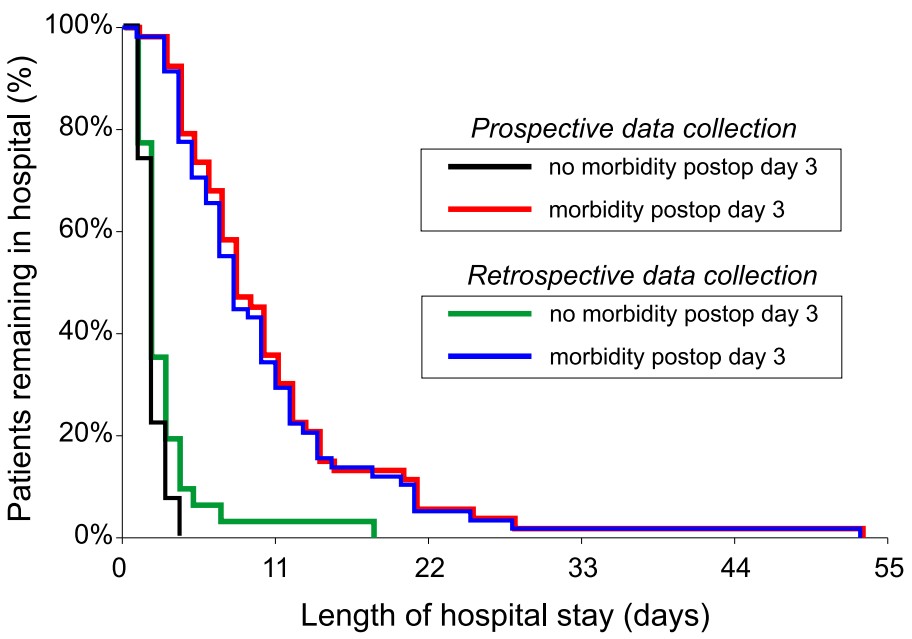

**Figure 2  Kaplan–Meier plot of percentage of patients remaining in hospital against time by length of hospital stay.** The plot demonstrates increased hospital length of stay in those sustaining POMS on postoperative day 3, as recorded by both retrospective (hazard ratio: 5.0 (95% CI [2.3–10.6]); $p < 0.001$, by log rank test) and prospective (hazard ratio: 3.6 (95% CI [2.0–6.8]); $p < 0.001$, by log rank test) modes of data collection.

**Table 3  Postoperative Day 7, and non-inferiority for sensitivity of mode of data collection.**

| Domain | P value | 95% CI | Non-inferior? |
|---|---|---|---|
| Any type | 0.0002 | −0.17–0.04 | Yes |
| Supplementary oxygen | 0.0008 | −0.088–0.088 | Yes |
| Antibiotics | 0.0569 | 0.0036–0.2305 | Yes |
| Temperature | 0.0004 | −0.13–0.065 | Yes |
| Urinary catheter | 0.0089 | −0.042–0.17 | Yes |
| PONV | 0.0008 | −0.088–0.088 | Yes |
| Enteral feeding | 0.0002 | −0.17–0.042 | Yes |
| Confusion | 0.0008 | −0.088–0.088 | Yes |
| Myocardial infarction | 0.0004 | −0.1288–0.065 | Yes |
| Arrhythmias | 0.0028 | −0.065–0.129 | Yes |
| Drain | 0.0008 | −0.088–0.088 | Yes |
| Packed red cells | 0.0004 | −0.13–0.065 | Yes |
| Products | 0.0008 | −0.088–0.088 | Yes |
| Pain | 0.0004 | −0.13–0.065 | Yes |
| Parenteral opioids | 0.0008 | −0.088–0.088 | Yes |
compared to that collected prospectively. POMS-defined morbidity on postoperative day 3 captured retrospectively and prospectively was linked with similarly delayed hospital discharge. The non-inferiority of retrospective data collection may provide significant economic benefit through reducing the need for research staff collecting data in real-time. Furthermore, bioinformatic tools capable of mining clinical datasets should enable time-stamped, highly granular information on postoperative morbidity. Our findings support the validity of future academic studies interrogating postoperative data using this standardized retrospective approach.

Postoperative complications are critical predictors of long-term mortality, irrespective of preoperative risk factors (*Khuri et al., 2005*). Defining surgical patients at risk of delayed adverse outcomes requires robust, sensitive measures of postoperative morbidity. Other systems such as the Clavien–Dindo scale are tremendously useful for measuring deviations from usual care and severity of postoperative morbidity. However, the Clavien–Dindo system, as with all clinical assessment models, has had various shortcomings highlighted (*Rassweiler, Rassweiler & Michel, 2012*). Discriminating between surgical errors and apparently unforeseeable complications is challenging. A urology study demonstrated that surgeons disagree widely on what constitutes a complication for Clavien–Dindo grading (*De la Rosette et al., 2012*).

There may be significant merit in combining systems to capture patient-centered outcomes across the spectrum of low–high risk surgery and also to reflect severity of complications, as recently utilized in a perioperative randomized controlled trial (*Ackland et al., 2015*). Irrespective of which system may be used, routine reporting of outcomes in noncardiac patient population is limited; registry data has chiefly focused on technical and procedural outcomes. POMS is an attractive tool as the survey questions are rapidly completed, have high inter-observer agreement and are patient-centered outcomes. However, POMS to date has only utilized prospectively collected data. The apparent need for prospective data collection not only makes larger comparisons of reported outcomes challenging, but also hard to implement on a widespread scale. This retrospective approach could therefore facilitate significant progress into expanding the number of patients for whom postoperative morbidity data can usefully be collected, with a substantial impact on clinical and translational research work as a result.

Our data is consistent with other studies where POMS-defined morbidity is present in up to 75% of patients is associated with prolonged hospital stay. Although the data represents a single-center case series, our post-hoc analysis of a well-validated, prospective descriptor of morbidity (POMS) is the first study of its kind. A strength of this study is that bias was minimized through data collection and retrospective analyses being performed by blinded independent investigators. We demonstrate that postoperative data recorded prospectively can be attained and analyzed using traditional systems of retrospective data collection. This certainly may reduce the costs of postoperative morbidity data collection, and suggest their incorporation into electronic patient records would be a surmountable software challenge. Length of hospital stay is increased by 'minor' postoperative complications (e.g., nausea and vomiting), which impacts on the financial

burden of medical healthcare. In this clinical setting, we have established a system to analyze data in a more cost-effective way, to tackle this problem.

## CONCLUSION

POMS-defined postoperative morbidity can be analyzed retrospectively. This approach suggests that these data can be easily captured from electronic patient record systems, thereby expanding the potential for bioinformatics approaches to generate new clinical and translational insights into postoperative recovery. In this population, the non-inferiority of retrospective data collection may contribute to real-time risk stratification, and warn of the true incidence and duration of postoperative complications.

### Funding

This project was funded in part by the British Journal of Anaesthesia/National Institute of Anaesthesia Basic Science Fellowship (GLA); National Institute for Health Research Central and East London Clinical Research Network (AR, GLA). This work was undertaken at University College London Hospitals NHS Trust/University College London who received a proportion of funding from the Department of Health UK NIHR Biomedical Research Centre funding scheme. The funders had no role in study design, data collection and analysis, decision to publish, or preparation of the manuscript.

### Grant Disclosures

The following grant information was disclosed by the authors:
British Journal of Anaesthesia/National Institute of Anaesthesia Basic Science Fellowship (GLA).
National Institute for Health Research Central and East London Clinical Research Network (AR, GLA).
Department of Health UK NIHR Biomedical Research Centre.

### Competing Interests

The authors declare there are no competing interests.

### Author Contributions

- Amour B.U. Patel conceived and designed the experiments, performed the experiments, analyzed the data, wrote the paper, prepared figures and/or tables, reviewed drafts of the paper.
- Anna Reyes conceived and designed the experiments, performed the experiments, analyzed the data, wrote the paper, reviewed drafts of the paper.
- Gareth L. Ackland analyzed the data, contributed reagents/materials/analysis tools, wrote the paper, prepared figures and/or tables, reviewed drafts of the paper.

## Human Ethics

The following information was supplied relating to ethical approvals (i.e., approving body and any reference numbers):

Medical Research Ethics Committee: 10/WNo03/25.

## Data Availability

The raw data is supplied in Supplemental Information 1.

## Supplemental Information

Supplemental information for this article can be found online at http://dx.doi.org/10.7717/peerj.1466#supplemental-information.

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
