# Peer review of "Non-inferiority of retrospective data collection for assessing perioperative morbidity"

_PeerJ, doi:10.7717/peerj.1466_

## Round 0.1 · original submission · Major Revisions

Based on the reports at hand, I request a major revision for further review. Please respond to the reviewer comments. The main contribution should be highlighted in the new version.

Reviewer 1 ·

Basic reporting

1. The value or the importance of this manuscript has not been fully elucidated. For example, the author said they tested by three different approaches, which included 1) to assess whether POMS-defined morbidity captured retrospectively or prospectively altered the trajectory of patients becoming free of postoperative morbidity , and 2) to assess whether POMS-defined morbidity on postoperative day 3 captured retrospectively or prospectively was linked with delayed hospital discharge. However, few (or not too much) discussion was related to these in the discussion. Additional discussions on this point might be helpful for proper understanding of the significance of the results.
2. Additional and more careful descriptions must be needed in the figure legend part corresponding to Figure 1 and 2.
3. Additional description of previous study should be needed. For example, in page 12, “However, the Clavien-Dindo system, as with all clinical assessment models, has had various shortcomings highlighted.(Rassweiler et al. 2012).” What shortcoming is Clavien-Dindo system ? Another example, “where POMS-defined morbidity is present in ~75% of patients”, it seems no similar results addressed in current manuscript.

4. I'm not sure, but I'm afraid line colors in figure 2 are wrong.

Experimental design

1. In the method, “Morbidity was recorded by a trained investigator as defined by POMS prospectively on postoperative days 3 and 7. Separately, an independent investigator…” only one investigator evaluated the whole material? How did you control the quality of the whole study?
2. The clinic data, such as the average age, gender and what kind of disease of the 85 patients should be mentioned shortly in the manuscript, or perhaps, authors can show them in the supplementary Materials.

Validity of the findings

1. Although the results showed that retrospective compilation of POMS-defined morbidity data from charts, medical and nursing notes was non-inferior to prospective, real-time data collection requiring research or administrative personnel. But in figure 1 and 2, there was a little difference (although no significance statistically) comparing retrospective and prospective modes of data collection, is there any meaning for this difference?

Additional comments

This manuscript describes retrospective data collection from routinely collated hospital data from paper and electronic charts, medical and nursing notes was non-inferior to prospective data collection requiring research staff capturing POMS-defined morbidity in real-time. I think the issue mentioned in this manuscript is very important and their approach is technically sound. Nevertheless, I think there is still room for further improvements of the manuscript.

Annotated reviews are not available for download in order to protect the identity of reviewers who chose to remain anonymous.

Reviewer 2 ·

Basic reporting

In this paper, the authors compared retrospective data collection with prospective data collection to predict postoperative morbidity. And they have shown that the method of retrospective data collection was non-inferior to prospective data collection when analyzing the real data.

Experimental design

1) Figure 1 and 2 show the results for patients free of POMS-defined morbidity comparing retrospective and prospective models of data collection. In the introduction, authors mentioned the real time data collection. How about the real time data collection? It would be better if comparing these three methods of data collection.

2) In the introduction, authors mentioned that they assessed whether POMS-defined morbidity captured retrospectively or prospectively at postoperative day 3 was linked with delayed hospital discharge. But in results section, there are no comparisons of two models. Please add the related results.

Validity of the findings

No Comments

Additional comments

In this paper, the authors compared retrospective data collection with prospective data collection to predict postoperative morbidity. And they have shown that the method of retrospective data collection was non-inferior to prospective data collection when analyzing the real data.
Some revisions are needed as follows:

1) Figure 1 and 2 show the results for patients free of POMS-defined morbidity comparing retrospective and prospective models of data collection. In the introduction, authors mentioned the real time data collection. How about the real time data collection? It would be better if comparing these three methods of data collection.

2) In the introduction, authors mentioned that they assessed whether POMS-defined morbidity captured retrospectively or prospectively at postoperative day 3 was linked with delayed hospital discharge. But in results section, there are no comparisons of two models. Please add the related results.

---

## Round 0.2 · accepted · Accept

The authors made point-by-point revision based on the reviewers. The concerns were largely addressed. The manuscript has been improved accordingly. Now I suggest its acceptance for publication.